# Computational Study of Stiffness-Tuning Strategies in Anguilliform Fish

**DOI:** 10.3390/biomimetics8020263

**Published:** 2023-06-16

**Authors:** Zuo Cui, Xuyao Zhang

**Affiliations:** 1School of Aerospace Engineering, Guizhou Institute of Technology, Guiyang 550003, China; 2School of Mechatronic Engineering, Guizhou University, Guiyang 550025, China; zhanggzu@163.com

**Keywords:** anguilliform fish, planar serial-parallel mechanism, stiffness-tuning strategies, swimming speed

## Abstract

Biological evidence demonstrates that fish can tune their body stiffness to improve thrust and efficiency during swimming locomotion. However, the stiffness-tuning strategies that maximize swimming speed or efficiency are still unclear. In the present study, a musculo-skeletal model of anguilliform fish is developed to study the properties of variable stiffness, in which the planar serial-parallel mechanism is used to model the body structure. The calcium ion model is adopted to simulate muscular activities and generate muscle force. Further, the relations among the forward speed, the swimming efficiency, and Young’s modulus of the fish body are investigated. The results show that for certain body stiffness, the swimming speed and efficiency are increased with the tail-beat frequency until reaching the maximum value and then decreased. The peak speed and efficiency are also increased with the amplitude of muscle actuation. Anguilliform fish tend to vary their body stiffness to improve the swimming speed and efficiency at a high tail-beat frequency or small amplitude of muscle actuation. Furthermore, the midline motions of anguilliform fish are analyzed by the complex orthogonal decomposition (COD) method, and the discussions of fish motions associated with the variable body stiffness and the tail-beat frequency are also presented. Overall, the optimal swimming performance of anguilliform fish benefits from the matching relationships among the muscle actuation, the body stiffness, and the tail-beat frequency.

## 1. Introduction

In the swimming cycles, most fish contract their muscles left and right rhythmically and propel forward by the undulations of the body and/or caudal fin. Now more and more studies show that swimming fish use muscular activities to adjust their body stiffness in real-time [1,2], and the property of various stiffness has significant influences on the thrust and/or swimming efficiency [3,4,5]. In early works, Wu [6] studied two-dimensional flexible hydrofoils and found that flexible hydrofoils could be more efficient than rigid ones at producing thrust. Several theoretical and experimental studies also confirmed that adding flexibility to a hydrofoil can improve thrust and/or efficiency [7,8,9]. Tytell et al. [10,11] developed both physical and computational models of lamprey fish and found that the bending stiffness of the fish body affected the wake structures behind the tail, as well as the swimming efficiency. McHenry et al. [12] also found that flexibility can be used to increase the forward speed of swimming fish. Additionally, several fish-like robots were developed to study the role of flexibility. For example, Jusufi et al. [13] used antagonistic pneumatic actuators to develop a robotic fish with variable body stiffness; White et al. [14] added flexible joints into a tuna-inspired robot to study the capability of high-frequency locomotion. 

From the biomechanical aspect, flexible fish would maximize their body deformations when actuated at their resonant frequency because of the phenomenon of resonance [15]. In the experiments of robotic lamprey [16], the results suggested that the resonance played an important role in maximizing efficiency, with an increased stride length but only a very small expansion of wake energy. By contrast, several studies of a flexible foil [17,18] found that the peaks in amplitude did not always correspond to peaks in thrust and/or efficiency. For example, Quinn et al. [2] found that the efficiency of robotic fish may peak at just 0.4–0.6 times the resonant frequency. Therefore, someone believes that the thrust and/or efficiency of swimming fish is maximized by changing the aerodynamic variables, such as the camber, the angle of attack, and the undulatory motions. For instance, the flexibility of the fish body created a passive phase offset between the actuation force and its response, and this phase offset could reduce the effective angle of attack at the leading edge of the tail/fin, which affected the formation of leading-edge vortices (LEVs) and leads edge suction, as well as the swimming efficiency [19,20]. Meanwhile, flexibility also affects the temporal evolution of fish propulsive motions, which can reorient forces in ways that boost thrust and reduce drag [21]. Overall, the efforts to refine the role of flexibility are still ongoing, and there is no clear explanation about the stiffness-tuning strategies in fish locomotion. 

Several dynamic models of swimming fish have been developed to investigate the influences of body stiffness. Williams et al. [22] used the viscoelasticity beam model to study the deformed patterns of the fish body, which was actuated by different types of muscle actuation. McMillen et al. [23,24] simplified anguilliform fish as an elastic rod and a revised muscle model was proposed to produce a more realistic actuation force. Moreover, Tytell et al. [1] built a neuromechanical model of lamprey fish to study the interactions between internal forces, body stiffness, and fluid environment. Porter et al. [25] developed a dynamic model to study the force transmission along the vertebral column of dogfish sharks.

By comparison, the dynamic model of swimming fish, proposed by McMillen et al. [24], is very suitable for studying the stiffness-tuning strategies of anguilliform fish. Different from the previous model, the redundantly actuated planar series-parallel structure is adopted to demonstrate the biological structure of the fish body, in which the stiffness can be adjusted as needed. We intend to expand the musculo-mechanical model to explore the influences of various body stiffness on swimming performance in the hope of gaining insight into the way it is performed in real fish. Some control variables, such as the strength and frequency of muscle activation, may be enlarged or narrowed as possible, and we will look for patterns that could give insight into the right matching relations among the muscle actuation, the swimming performance, and the variable body stiffness. For example, we will measure the swimming speeds at different frequencies for a given level of muscle activation or at different levels of muscle activation for a given body stiffness. Although how fish use active muscle tensioning to tune their body stiffness to improve thrust and/or efficiency in real-time cannot be fully explained at the moment, we try to throw light on this problem in the present study.

The stiffness-tuning strategies in anguilliform fish can be implemented in next-generation fish-like robots or underwater vehicles, with an immediate improvement in forward speed and propulsive efficiency. The stiffness-tuning mechanisms and automatic stiffness control based on various swimming conditions are also promising applications. Moreover, while the stiffness-tuning strategies are poised to improve fish-like robots, they would offer insights into new studies of live fish locomotion, with the requirements of new challenging experiments, such as invasive surgery with implanting transducers and electrodes into the fish body, vivo stiffness measurements of various fish species, etc. The rest of this paper is organized as follows. The dynamic model of anguilliform fish is presented in Section 2. The computational results are demonstrated in Section 3, and the influences of various body stiffness on swimming performance are analyzed in Section 4. The discussions and conclusions are presented in Section 5 and Section 6, respectively.

## 2. Dynamic model of Anguilliform Fish

### 2.1. Series-Parallel Structure of Fish Body

In nature, lamprey fish have elongated bodies with narrow margins of caudal fins, and they undulate like a flapping ribbon. In the present study, the geometry of the fish body is measured from a real lamprey fish [26]. The cross-section shape of the fish body is approximated as elliptic, and the major and minor axes are labeled by $a\left(x\right)$ and $b\left(x\right)$, respectively.
(1)$$a\left(x\right)=\left\{\begin{array}{c} \sqrt{0.08Lx-x^{2}},\;0\leq x<0.04L\;\\ 0.04L-0.03L{\left(\frac{x-0.04L}{0.91L}\right)}^{2},\;0.04L\leq x\leq 0.95L\\ \left(L-x\right)/5,\;0.95L<x\leq L\; \end{array}\right.$$
(2)$$b\left(x\right)=0.08L\sqrt{1-{\left(\frac{x-0.51L}{0.51L}\right)}^{2}}$$

In Equations (1) and (2), L is the full length of the fish body. In general, the motions of anguilliform fish occur only in a horizontal plane, and a two-dimensional fish model with constant height is acceptable. As shown in Figure 1, a planar serial–parallel mechanism with variable stiffness is proposed to model the biological structure of anguilliform fish. 

As displayed in Figure 2, the body of anguilliform fish is modeled by a chain of interconnected planar serial-parallel structure units, which consists of rigid links, two springs, two dampers, and a revolute joint. The revolute joints link each vertebra into a continuing spine, representing the flexible notochord. The muscles are attached to the rigid vertebrae, which are modeled by a parallel structure of spring and damper. When the muscles are elongated or contracted, the actuation is imposed along two sides of the fish body, and the whole serial–parallel structure deforms like a flexible fish body. Additionally, the geometry parameters of real fish are adopted to design the serial–parallel structure, and the muscle actuations are determined by the muscle length and the velocity of its shortening.

The working principle of a single rotational serial–parallel unit is shown in Figure 2. The top platform A_1_A_3_A_2_ is supported by the middle rigid leg OB_3_ and the elastic legs A_1_B_1_ and A_2_B_2_. The elastic legs A_1_B_1_ and A_2_B_2_ on both sides connect the rotating pairs of upper revolute joints A_1_ and A_2_ on the top platform and the rotating pairs of lower revolute joints B_1_ and B_2_ on the fixed platform B_1_B_3_B_2_. The rigid middle leg OB_3_ is fixed on the fixed platform. The top platform rotates around the rotating center with a single degree of freedom. The muscle actuations change the internal forces resulting from the elastic legs A_1_B_1_ and A_2_B_2_, and the internal forces balance each other in the closed mechanism. The ribs A_1_A_3_ (or B_1_B_3_) and A_2_A_3_ (or B_2_B_3_) have the same length. The longitudinal length h is the spatial distance between the adjacent serial–parallel units.

For the discrete fish model, the dynamic equations of ith serial–parallel unit are expressed as
(3)$$\left\{\begin{array}{c} m_{i}a_{i}=hW_{i}+\sum F_{i},\\ J_{i}{\ddot{\psi }}_{i}=\sum M_{i}+\sum L_{i}\times F_{i}, \end{array}\right.\;$$
where $m_{i}$ is the mass of ith serial–parallel unit; $a_{i}$ is the acceleration of serial–parallel unit; $hW_{i}$ is the hydrodynamic force applied on the ith serial–parallel unit; $F_{i}$ is the inner force transmitted from the adjacent structure unit; $L_{i}$ is the arm of force $F_{i}$. $J_{i}$ is the inertia moment of serial–parallel unit; $M_{i}$ denotes the bending moment generated by muscles. 

In Equation (4), the hydrodynamic force W is calculated by Taylor’s resistance model [27], and the components in the normal direction $\vec{n}$ and the tangential direction $\vec{\tau }$ can be written as: (4)$$\left\{\begin{array}{c} W=W_{n}n+W_{\tau }\tau \\ W_{n}=-\left(a\rho_{f}\left|v_{n}\right|v_{n}+\sqrt{8\rho_{f}a\mu_{f}\left|v_{n}\right|}v_{n}\right)\\ W_{\tau }=-2.7\sqrt{2\rho_{f}a\mu_{f}\left|v_{n}\right|}v_{\tau } \end{array}\right.\;$$
where $v_{n}$ and $v_{\tau }$ represent the normal and tangential velocity along the surface of the fish body, respectively; $\rho_{f}$ is the fluid density; $\mu_{f}$ is the fluid viscosity.

### 2.2. Force Analyses

In Figure 2, the bending fish body is modeled by the planar serial-parallel mechanism, in which the joints connecting each serial-parallel unit of length h is actuated by a pair of spring-dashpot-actuators in parallel, anchored to arms of length w that project normally from the links’ midpoints. These arms represent myosepta, the connective tissue layers to which the muscle fibers connect. The linear springs and dashpots represent the tissue’s passive viscoelasticity, described by the spring constant k and the damping coefficient γ. The actuation forces $f_{Li}$ and $f_{Ri}$ are generated by the contractile muscles on the right and left sides of ith planar serial-parallel unit, respectively. Suppressing the dependence on i and denoting the relative extensions $\Delta_{R}$ and $\Delta_{L}$, the total forces on the right and left sides can be written as
(5)$$G_{R}\left(t\right)=f_{R}\left(t\right)+k\Delta_{R}+\gamma \dot{\Delta_{R}}$$
(6)$$G_{L}\left(t\right)=f_{L}\left(t\right)+k\Delta_{L}+\gamma \dot{\Delta_{L}}$$
when $\Delta_{R}>0$, $\Delta_{L}>0$, the springs are in tension, hence generating contractile forces. The forces are applied at a distance w from the centerline of the fish body, so elementary trigonometry gives:(7)$$\Delta_{R}=\frac{\bar{CD}-h}{h}=\frac{2dcos\left(\alpha^{\prime }\right)-h}{h}=cos\left(\frac{\psi_{i}}{2}\right)-1-2\frac{w}{h}sin\left(\frac{\psi_{i}}{2}\right)$$
(8)$$\Delta_{L}=\frac{\bar{AB}-h}{h}=\frac{2dcos\left(\alpha \right)-h}{h}=cos\left(\frac{\psi_{i}}{2}\right)-1+2\frac{w}{h}sin\left(\frac{\psi_{i}}{2}\right)$$
where $\psi_{i}=\varphi_{i+1}-\varphi_{i}$ is the angle between neighboring links. Finally, the moment arms $L_{R}$, $L_{L}$ to the joint along normal from the lines A_1_B_1_ and A_2_B_2_ on which the forces act are computed as
(9)$$L_{R}=d\sin\left(\alpha^{\prime }\right)=d\cos\left(\beta -\psi_{i}/2\right)=wcos\left(\frac{\psi_{i}}{2}\right)+\frac{h}{2}sin\left(\frac{\psi_{i}}{2}\right)$$
(10)$$L_{L}=d\sin\left(\alpha \right)=d\cos\left(\beta +\psi_{i}/2\right)=wcos\left(\frac{\psi_{i}}{2}\right)-\frac{h}{2}sin\left(\frac{\psi_{i}}{2}\right)$$

For the ith serial–parallel unit with small rotation angles, the resulting torque at the joint is given by
(11)$$M_{i}=G_{Ri}L_{Ri}-G_{Li}L_{Li}\;\approx \left[f_{Li}\left(t\right)-f_{Ri}\left(t\right)\right]w+\left[2\upsilon w^{2}-\frac{h^{2}}{4}\left(f_{Li}\left(t\right)+f_{Ri}\left(t\right)\right)\right]\left(\frac{\varphi_{i+1}-\varphi_{i}}{h}\right)+2\gamma w^{2}\left(\frac{{\dot{\varphi }}_{i+1}-{\dot{\varphi }}_{i}}{h}\right)$$

The forces $\;f_{Li}$ and $f_{Ri}$ are calculated by the calcium kinetics model (see Section 2.3). To approximate a uniform distribution of the muscle, we set $w\left(x\right)=b\left(x\right)/2$, $b\left(x\right)$ is the half-width of the fish body, as described in Equation (2). According to the muscle model, the generated actuation force is a function of muscle fibers; therefore, we propose that the stiffness k and damping γ are proportional to the cross-sectional area of the fish body. Thus we set
(12)$$k=ab\bar{k},\gamma =ab\bar{\gamma }$$
where $\bar{k}$ and $\bar{\gamma }$ are the stiffness and damping of muscle fibers, with the units N/m^2^ and N s/m^2^, respectively. In particular, Young’s modulus E is used to describe the body stiffness, which is written in terms of the spring stiffness as $E=2\bar{k}/\pi$. 

### 2.3. Muscle Activation Model

According to the study of Hill [28], each myotome is modeled as a contractile element (CE) in series with an elastic element (SE). The CE and SE are located in series and experience equal forces at steady-state. The active force is generated by the CE, while the positive force is exerted by the SE, which is modeled as a linear spring. In reality, the stretch of the SE is not instantaneous due to the activation of the CE. The actuation force $f_{R,L}$ is dependent upon both the muscle length and its shortening velocity. The force density is proportional to muscle activation, scaled by the amplitude factor D that represents the force per unit muscle cross-section generated by unit activation: (13)$$f_{R,L}=\chi_{R,L}\left(t\right)D$$

In Equation (14), $\chi_{R,L}\left(t\right)\;$ refers to muscle activation, and the subscripts R and L indicate the right and left side, respectively. The muscle activation $\chi \left(t\right),$ in turn, produces contractile forces in the muscle fibers. In the present study, the model of calcium kinetics is adopted to describe the muscle activation $\chi_{R,L}\left(t\right)$, and it has been revised by the experiments of single myotomes of fish muscle [23]. The parameters $k_{1}\sim k_{4}$ of calcium, the kinetics model is the same as the data in [24], and the intrinsic shape determined by the model of calcium kinetics is illustrated in Figure 3. The results show that the actuation force agreed well with the experimental data, and it is very close to the muscle force in refs. [22,23]. The results also suggested that the calcium kinetics model can accurately simulate muscle activation and force generation that occur during fish swimming.

In anguilliform fish, the musculature consists of approximately 100 myotomes, which are activated sequentially from just behind the gills towards the tail, alternating on the left and right sides of the body [22]. During steady swimming, the relative timing between muscle activation and body curvature is shown in Figure 4. The dashed lines along the fish body show the regions of muscle activation. The solid bars represent the electromyogram (EMG) activity measured at four different points along the left side of the body, plotted against time, and the sinusoidal lines are a measure of the local curvature of the body, which gives a measure of muscle length. This relative timing in Figure 4 is illustrated as a changing phase delay between the beginning of muscle stimulation and shortening. The relative timing between activation and curvature is considered in the present muscle model, which allows a larger frequency of activation, different from that in McMillen et al. [24]. Additionally, we change the amplitude factor D to study the influences of muscle actuation.

### 2.4. Solution of Fish Dynamic Model

In the fish dynamic model, the body is divided into N sections, and a chain of serial-parallel units with each of length h, with mass $m_{i}$ at each pivot, is redundantly actuated by active muscle forces. If the entire fish body is actuated, 6 (N−1) first-order ordinary differential equations (ODEs) describe the muscle forces in the N-link chain, and with the 3N second order ODEs, they jointly determine the body dynamics. The Newton iterative scheme, the spatial and temporal discrete methods provided in reference [23] (Appendix (2)∼(4) in [23]), are adopted to solve the dynamic equations. The boundary conditions at the fish head and tail is free, i.e., the force and moment are zero. The flow chart to solve the finite-difference discretization equations of the fish model is presented in Figure 5. The algorithm of ode45 in the Matlab software (Matlab R2022a) is used to solve the numerical equations, and the time step is 0.001 s.

## 3. Computational Results of Fish Undulatory Motions

### 3.1. The Variables of Fish Model

In the fish model, the shape of anguilliform fish is described by Equation (2). The body length is 20 cm, and it is discrete into 40 segments. The passive properties of the fish body are described by the density ρ, Young’s modulus E and the viscosity μ. The water viscosity $\mu_{f}$ is 10^−3^ Pa·s, and the density $\rho_{f}$ is 1 g/cm^3^. The density of fish bodies is as same as that of water. Real fish have complex networks of muscle, cartilage, bone, skin, and organs, and they are more complicated than any physical fish model. Therefore, it is necessary to simplify anguilliform fish as a rod with heterogeneous viscoelastic materials, which can be described by the constant Young’s modulus and viscosity coefficient.

In the study of eel fish [29], Young’s modulus is 0.39 MPa, and the viscosity coefficient is 11 kPa·s. In Ref. [30], Young’s modulus of hagfish is 0.30 MPa, and the viscosity coefficients are 2 kPa·s and 5.7 kPa·s when the tail-beat frequencies are 3 Hz and 1 Hz, respectively. Therefore, it is reasonable to set the initial Young’s modulus E as 0.1 MPa, and the initial viscosity μ as 3 kPa·s. The fish head, occupying 1/10 of the body length, is without activation muscles, while the remaining 9/10 of the fish body that is capable of activation works as the activation region. To study the influence of various body stiffness, the Young’s modulus is varied in the possible range, and the variables of the fish model are listed in Table 1. The nomenclature of all parameters in the fish model is listed in Appendix A. 

### 3.2. Swimming Performance of Anguilliform Fish

In our simulations, anguilliform fish starts from rest and accelerates to a quasi-steady state with maximum forward velocity along a straight way. The lateral and forward velocities of the virtual fish are displayed in Figure 6. The solid red line is the forward speed, and the dashed curve is the averaged forward speed over the tail-beat period. The dotted line is the lateral speed at the mass center of the fish body, with a sine wave with an oscillating frequency of 1 Hz. The mean asymptotic speed (MAS) is about 0.60 BL/s (body lengths per second). Besides, we divide the fish body into 100 segments and compare the resulting forward speeds. The results show that the speed obtained from a body of 40 sections differs by less than 0.1% from that of 100 sections.

According to the large-amplitude elongated body theory [31], swimming efficiency can be calculated by
(14)$$\eta =\frac{1}{2}\left(1+\frac{U}{V}\right)$$
where U is the forward speed, and V is the wave speed. In the present case, the wave speed is 1.05 BL/s, and thus the swimming efficiency is 78.6%. The successive shapes of anguilliform fish in one tail-beat cycle are shown in Figure 7. The simulations readily yield results that are qualitatively similar to the movements of real fish. Color intensity on the fish body indicates the absolute value of the active bending moment. The red region shows the propagation curve of the active moment crest, while the blue line shows the propagation curve of the positive moment crest. The red and blue dashed lines near two sides of the body indicate the positive and negative curvature regions, respectively.

During one tail-beat cycle, the tracings of swimming fish that approximate its body centerline are demonstrated in Figure 8. The oscillating amplitude of the fishtail is obviously larger than that of the fish head. The undulatory motions in Figure 8a result from the dynamic fish model, which is similar to the tracings in Figure 8b from real fish [32]. These results demonstrate that the propagating wave can naturally emerge in a forced planar serial-parallel structure with redundant muscle actuations, which can be used to study the influences of various body stiffness.

### 3.3. Complex Modal Characteristics of Body Motions

As shown in Figure 8, it is very difficult to distinguish the undulatory motions of fish bodies accurately. In our previous study [33], the COD (complex orthogonal decomposition) method is adopted to describe the differences in the undulatory motions, including the oscillating amplitude of the fish head, the tail’s attack angle, the wavelength. For the integrity of this study, the COD method is briefly introduced. At first, the midline motions of fish body $y\left(x,t\right)$ are dispersed to finite elements $y\left(x_{m},t_{n}\right)$, $x_{m}=mL/M\left(m=1,\;2,\;\ldots \ldots ,\;M\right)$ is different positions along the fish’s body, and $t_{n}=nT/N\;\left(n=1,\;2\ldots \;N\right)$ is different time instants in one tail-beat period T. Secondly, a column vector **y_m_**(**t**) is constructed as [y_m_(t_1_) y_m_(t_2_) y_m_(t_3_)…y_m_(t_n_)]^T^, and then the Hilbert transform is used to transform **y_m_**(**t**) into the complex signals **z_m_**(**t**):**z_m_**(**t**) = **y_m_**(**t**) + i**H**(**y_m_**(**t**)), i^2^ = −1.(15)
where **H**(**y_m_**(**t**)) is the Hilbert transform of **y_m_**(**t**)**.** At last, a complex matrix $Z_{M\times N}=$ [**z_1_**(**t_n_**), **z_2_**(**t_n_**), **……**, **z_m_**(**t_n_**)]^T^ is constructed, and the eigenvalues $\lambda_{m}\;$ and eigenvectors $w_{m}$ of matrix $Z_{M\times N}$ are solved. The traveling index ξ is defined by
(16)$$\xi =1/\mathrm{cond}(\left[\mathrm{real}\left(w_{1}\right),\mathrm{imag}\left(w_{1}\right)\right]$$
where **cond**() is the operating symbol of the condition number; $w_{1}$ is the first mode of the eigenvectors $w_{m}$; $\mathrm{real}\left(w_{1}\right)$ is the real part of $w_{1}$, and $\mathrm{imag}\left(w_{1}\right)$ is the imaginary part of $w_{1}$. The traveling index is used to describe the independence between the real and imaginary parts. When the real and imaginary vectors are dependent, the motions are pure standing waves, and the traveling index is 0; when the real and imaginary vectors are completely independent, the motions are pure traveling waves, and the traveling index is 1.

In the COD method, the undulatory motions are decomposed into two parts: pure standing waves and pure traveling waves, and the traveling index is defined by the ratio between the parts of the standing wave and the traveling wave. The midline motions in Figure 8 are decomposed into two parts, pure standing waves, and pure traveling waves, as shown in Figure 9, and the traveling index is 0.672. The results indicate that the midline motions of anguilliform fish are mixed by standing waves and traveling waves.

## 4. Swimming Performance of Anguilliform Fish

### 4.1. Fish Body with Fixed Body Stiffness

Biological evidence showed that the range of muscle activation was 0.1% to 20% of maximum isometric strength [22], so the magnitude of muscle activation is changed by timing an actuation coefficient D, which is also varied from 0.01 to 0.2. In this section, the swimming performance of anguilliform fish is studied when the stiffness of the fish body is fixed, and four different Young’s moduli, 0.006 MPa, 0.06 MPa, 0.6 MPa, and 1.0 MPa, are selected in this section. The relations among the tail-beat frequency, the amplitude of muscle actuation, and the swimming speed are shown in Figure 10. The tail-beat frequency is increased from 0.5 Hz to 5 Hz in the interval of 0.5 Hz, and the results of swimming fish with fixed body stiffness are showed as below:(1)For fixed body stiffness, the forward speed is increased with the tail-beat frequency and then decreased from the maximum value slowly. The maximum speeds are varied with the actuation amplitude and the tail-beat frequency. For the cases in Figure 10a, Young’s modulus of the fish body is 0.006 MPa, when the actuation coefficients are 0.03 and 0.18, the maximum swimming speeds are 0.5 BL/s and 1.2 BL/s, and the corresponding tail-beat frequency is 1 Hz and 2.5 Hz, respectively. As a whole, for different fixed body stiffness, the maximum speeds are increased with the amplitudes of muscle activation.(2)For different fixed body stiffness in Figure 10, the maximum forward speed is also increased with Young’s modulus of the fish body. For example, when the amplitude of muscle activation D is 0.03, the maximum speeds are 0.5 BL/s in Figure 10a (E = 0.006 MPa, f = 1 Hz) and 1.3 BL/s in Figure 10b (E = 0.06 MPa, f = 3 Hz), respectively. It is notable that when Young’s modulus of the fish body is small, the amplitude of muscle actuation has a more significant influence on the swimming speed. The influence would be weak, with the increase of the muscle actuations and body stiffness, in particular, when f ≥ 3 Hz, E ≥ 0.06 MPa.(3)As to the muscle actuation amplitude, it has a complex relationship to the swimming speed, which depends on Young’s modulus of the fish body and the tail-beat frequency. In Figure 10a, when the tail-beat frequency is smaller than 1.5 Hz, the influence of muscle actuations on the swimming speed is weak, while the influence is strengthened with the increase of tail-beat frequency. By comparison, the influence of the amplitude of muscle actuations on the swimming speed in Figure 10c is strong when the tail-beat frequency is smaller than 3 Hz, while it becomes weak with the increase of tail-beat frequency. Another interesting finding in Figure 10c is that if the tail-beat frequency f ≥ 3 Hz, the swimming speed is increased with the amplitude of muscle actuations, but if f < 3 Hz, the swimming speed is decreased with the amplitude of muscle actuations. This phenomenon is also appeared in Figure 10a,b,d, although the critical frequency varies.

With different muscle actuations, the influences of tail-beat frequency on swimming efficiency are shown in Figure 11. Similar to the results in Figure 10, the swimming efficiency is also closely related to the tail-beat frequency, the body stiffness, and the amplitude of muscle actuation. For most cases in Figure 11, The swimming efficiency is firstly increased to the maximum and then decreased with the tail-beat frequency. The maximum swimming efficiency is also determined by the body stiffness, the muscle actuation, and the tail-beat frequency, and the changing pattern is similar to that of maximum swimming speed. But for the small actuation, i.e., D = 0.03, the swimming efficiency is decreased with the tail-beat frequency. As a whole, the swimming efficiency of anguilliform fish with small Young’s modulus (E = 0.006 MPa in Figure 11a) is smaller than that with large Young’s modulus in Figure 11b–d. Besides, when Young’s modulus of the fish body is larger than 0.06 MPa, the effects of muscle activation amplitude on the swimming efficiency is smaller, while the tail-beat frequency has a larger effect.

Combined the results in Figure 10 and Figure 11, for anguilliform fish with fixed body Young’s modulus, when the muscle activation amplitude is small (D = 0.03 N/m^3^ in the present study), the dynamic model of anguilliform fish is under-actuated and the forward speed and swimming efficiency are decreased with the tail-beat frequency. When the muscle activation is strengthened, the fish body is actuated appropriately, and as the increases of tail-beat frequency, the maximum efficiency and the maximum speed can be obtained. It is notable that the maximum efficiency is appeared at first, and then the maximum speed is followed. For example, when D = 0.05 N/m^3^ and E = 0.006 MPa, the maximum swimming efficiency is reached at f = 1 Hz, while the maximum forward speed is obtained at f = 1.5 Hz. Further, when the fish body is over-actuated, we find that with the increase of tail-beat frequency, both the forward speed and swimming efficiency cannot be improved anymore.

### 4.2. Fish Body with Variable Body Stiffness

#### 4.2.1. Influences on the Swimming Performance

In this section, Young’s modulus of the fish body is ranged from 10 kPa to 1 MPa, and the influences of variable body stiffness on swimming speed are shown in Figure 12. In Figure 12a, the tail-beat frequency is 1 Hz. For the cases D = 0.03 and 0.05, the swimming speed is increased with Young’s modulus of the fish body. However, for other cases, D ≥ 0.07, and the swimming speed is decreased with Young’s modulus of the fish body. In Figure 12b, the tail-beat frequency is 2 Hz, and for all the cases with different actuation amplitudes, the swimming speed is increased with Young’s modulus of the fish body. Further, we find that: (1) if E≤0.01 MPa, the swimming speed is proportional to the actuation amplitude; (2) if E>0.1MPa, the swimming speed is inversely proportional to the actuation amplitude (except for D = 0.03). In Figure 12c–e, the forward speed is increased with both the actuation amplitudes and Young’s modulus of the fish body. It is notable that under the same actuation, the forward speed of anguilliform fish is decreased with the tail-beat frequency. For example, for the cases with D = 0.15, when the tail-beat frequency is 3 Hz, 4 Hz, and 5 Hz, the swimming speeds are 1.5 BL/s, 1.3 BL/s, and 0.8 BL/s, respectively.

As shown in Figure 13, the swimming efficiency of anguilliform fish is also varied with Young’s modulus of the fish body. The changing patterns of swimming efficiency are similar to the results of swimming speed in Figure 12. For the tail-beat frequency is 1 Hz in Figure 13a, when D = 0.03 or 0.05, the swimming efficiency is increased with Young’s modulus of the fish body, reaching a steady state. However, for other actuation amplitudes, D ≥ 0.07, the swimming efficiency is decreased with Young’s modulus of the fish body. For the tail-beat frequency 3 Hz, 4 Hz, and 5 Hz, the swimming efficiency is increased with Young’s modulus and the muscle actuation, as shown in Figure 13c–e. The differences among these figures include: (1) with the fixed actuation amplitude and Young’s modulus of the fish body, the swimming efficiency is highest at f = 3 Hz and then decreases with the tail-beat frequency. (2) for the case of 3 Hz, the swimming efficiency is not increased with the actuation amplitude when D ≥ 0.11. This phenomenon also appeared in Figure 13d,e.

#### 4.2.2. Influences on the Midline Motions

The midline motions of anguilliform fish are also varied with Young’s modulus of fish body and tail-beat frequency, and they are investigated by the COD method. Meanwhile, the swimming speed, efficiency, and traveling index are used to evaluate the propulsive abilities of anguilliform fish. The results are presented in Figure 14 and Figure 15.

In Figure 14, the amplitude of muscle force D is 0.01, and the tail-beat frequency is f = 3 Hz. The changing pattern of swimming speed can be divided into three zones: low-speed zone, speed-increasing zone, and high-speed zone. In each zone, the typical Young’s modulus of the fish body, 0.004 MPa, 0.02 MPa, and 0.4 MPa, are selected to demonstrate the influences of variable stiffness on the midline motions. The undulatory motions of swimming fish are decomposed by the COD method, which is evaluated by the traveling index, as described in Section 3.3. As shown in Figure 14, although Young’s modulus of the fish body is varied from 0.004 MPa to 0.4 MPa, the basic pattern of undulatory motions are similar, i.e., the oscillating amplitude of the fish head and the tail are larger than that in the middle part. For the case of E = 0.004 MPa, the fish model is under-actuated, and the envelope amplitude of midline motions is smaller than in other cases (E = 0.02 MPa and 0.4 MPa). As a result, both the swimming speed and the swimming efficiency are lower, and the traveling index is also small. For the cases of E = 0.02 MPa and E = 0.4 MPa, the envelope amplitudes of midline motions are large enough, but there exist some differences, such as: (1) the oscillating amplitude of fish head is larger in the case of E = 0.02 MPa; (2) when E = 0.02 MPa, the lowest point of the envelope curves is located at about 0.2 *L* from the fish head, while there is no lowest point in the case of E = 0.4 MPa. Therefore, it is reasonable to predict that the large Young’s modulus of the fish body is easily prone to small head oscillations, and the envelope curviness of midline motions has a significant influence on the swimming efficiency.

The traveling index of midline motions is also varied with the tail-beat frequency. As shown in Figure 15, the amplitude of muscle force D is 0.01, and Young’s modulus of the fish body is fixed at 0.02 MPa. Three typical tail-beat frequencies, 1 Hz, 3 Hz, and 5 Hz, are selected to present their effects on the midline motions. When the tail-beat frequency is 1 Hz, it also means that the muscle activation frequency is relatively slow (as same as 1 Hz), which results in a large envelope amplitude of undulatory motions. The swimming efficiency is 80%, and the swimming speed is 0.62 BL/s. By comparison, when the tail-beat frequency is 3 Hz, the swimming performance is similar to the case of 1 Hz, although the envelope amplitude becomes small. However, for the case of 5 Hz, the muscle activation frequency is very fast, and the maximum amplitude of muscle activation (as shown in Figure 3) cannot be reached. The muscle activations cannot be fully used due to lacking enough reaction time, and thus the swimming efficiency is very low. Meanwhile, the traveling index is also very small.

## 5. Discussions

### 5.1. The Properties of Various Body Stiffness

To demonstrate the properties of various body stiffness, two different situations of anguilliform fish are selected: (A) Fixed body stiffness, E = 0.002 MPa, (B) Various body stiffness, Young’s modulus of the fish body ranged from 0.002 MPa to 1 MPa. Except for the body stiffness, the tail-beat frequency and the muscle actuations are the same. In detail, the tail-beat frequency is ranged from 0.5 Hz to 5 Hz, and four muscle actuations with different amplitudes, D = 0.03, 0.07, 0.11, and 0.15, are chosen. The comparisons of swimming speed and swimming efficiency are shown in Figure 16 and Figure 17, respectively.

Compared the swimming speeds of anguilliform fish with or without various body stiffness in Figure 16, we find that the most remarkable changes in the speed increment are appeared at the muscle actuation D = 0.03, while the mean increments of swimming speed are minimum when D = 0.15. Under certain muscle actuations, the speed increments are high when the tail-beat frequency is larger than 2.0 Hz. One distinctive feature is that when the muscle actuation is small, the influences of variable body stiffness are more obvious at lower tail-beat frequency, for example, in the cases of 1.5 Hz in Figure 16a,b. These results suggest that anguilliform fish may phone to vary their body stiffness to improve their swimming speed when the tail-beat frequency is large and/or the amplitude of muscle actuation is small.

For anguilliform fish with various body stiffness, the increments of swimming efficiency are shown in Figure 17. Similar to the results in Figure 16, for certain muscle actuation, the increments of swimming efficiency are large when the tail-beat frequency is between 1.5 Hz to 3.5 Hz, while the gap between cases A and B becomes small with the continuing increase of the tail-beat frequency. In Figure 17b–d, it is notable that the influence of variable stiffness on swimming efficiency is rather weak when the tail-beat frequency is lower than 1 Hz. Therefore, it is more effective for anguilliform fish to improve their swimming efficiency by varying the body stiffness at a certain range of tail-beat frequency. Additionally, more work should be performed to study the relationship between swimming efficiency and variable body stiffness because there are many approaches to calculating swimming efficiency.

### 5.2. The Effects of Hydrodynamic Forces

The hydrodynamic forces of anguilliform fish are modeled by the Taylor resistance theory in the present study. In this section, two different cases, with or without considering the hydrodynamic force, are selected to study the influences of the fluid force on the midline motions. In these cases, the amplitude of muscle actuation D is 0.01, and the tail-beat frequency is 1 Hz. Figure 18 shows the midline motion of anguilliform fish in a frame in which the center of mass is fixed. Different Young’s modulus E = 0.01 MPa, 0.05 MPa, 0.09 MPa are shown in three rows, and the left and right panels show the midline motions of the fish body with or without fluid forces, respectively. As a whole, the small curvature amplitude yields smaller motions in the central portions of the fish body, while the tail experiences smaller changes in curvature and tends to act as a ‘paddle’. The results show that the oscillation amplitudes of fish head and tail are obviously smaller than those without water force, which is consistent with the biology observations of anguilliform fish in reference [23]. Additionally, the results also show that the inclusion of hydrodynamic forces has a significant effect on the envelope shape of midline motions, although Young’s modulus is varied from 0.001 MPa to 0.09 MPa.

Further, the inclusion of hydrodynamic force also contributes to enlarging the traveling index. In our previous study [33], we collected 17 samples of anguilliform fish to analyze the range of traveling index, and the biological range is about 0.74~0.90. In present simulations, the range of the traveling index is 0.44–0.68, which is lower than the biological data. Besides, we compare the midline motions produced from fish dynamic model with those from real fish, and find that anguilliform fish deform its entire body with larger magnitude in nature, and the oscillation amplitude of the fish head is minimum. Therefore, it is reasonable to predict that the large oscillation amplitude of fish heads is harmful to the propulsive performance of anguilliform fish, and it also has a notable impact on the traveling index of midline motions.

From the perspective of dynamic analyses, the swimming fish can be regarded as a deformed viscoelastic rod immersed in water, and the deformation patterns of the fish body are dominated by their passive viscoelastic properties. Based on the viscoelasticity beam theory, the solution of this dynamic system can be a pure standing wave when the viscosity is zero (the dynamic model is a proportional damping system), and it could be a fish-like wave (a mixed standing and traveling wave) if the dynamic model is a non-proportional damping system. The strong damping modifies the system impedance and leads to an imbalance between the incident and reflected waves. This phenomenon was also proved by a series of bio-locomotion experiments [34], in which the transition mechanism from standing wave to traveling wave was analyzed. The wave propagation transforming from a standing wave to a traveling wave can be properly evaluated by the traveling index. That is the reason why we choose the traveling index to study the inner relations between the dynamic parameters and the undulatory motions of swimming fish. Therefore, the deformed patterns of anguilliform fish are depended on the stiffness, viscosity, and inertial mass. In the present study, the viscosity of a fish body is 1.0 kPa·s, so the influences of body viscosity and the interactive mechanism of muscle activation should also be considered in further work. In addition, we predict that both the stiffness and viscosity of the fish body are adjusted as needed, and these adjustments may be helpful to get more realistic midline motions of swimming fish.

## 6. Conclusions

In the present study, the stiffness-tuning strategies of anguilliform fish are studied by developing a dynamic model of swimming fish, with considerations of the morphological parameters, the muscle actuations, and the hydrodynamic forces. The properties of variable body stiffness have been analyzed in terms of its complex relations to muscle actuation, forward speed, and swimming efficiency. Moreover, the midline motions of swimming fish are decomposed by the COD method, and the traveling index is used to evaluate the differences in undulatory motions. The main conclusions are summarized below:(1)The maximum swimming speed and efficiency of anguilliform fish can be obtained by matching the body stiffness, the tail-beat frequency, and the muscle actuations appropriately. In general, when the body stiffness of a fish body is fixed, the swimming speed is increased to the maximum value and then decreased with tail-beat frequency. The maximum speed is increased with the amplitudes of muscle actuation and Young’s modulus of the fish body. The changing patterns of swimming efficiency are similar to that of swimming speed.(2)For the stiffness-turning strategies, anguilliform fish tend to vary their body stiffness to improve the swimming speed and efficiency at high tail-beat frequency. However, the influences of variable stiffness on swimming speed and efficiency are weakened when continuing the increase of tail-beat frequency. As to the swimming efficiency, it is also effective to vary their body stiffness when anguilliform fish are activated by the small amplitude of muscle actuations.(3)The midline motions produced by the dynamic model are similar to those of anguilliform fish observed from nature, but the range of the traveling index is lower than the biological range. It is partly because of the large oscillations of fish heads, and we also find the inclusion of hydrodynamic force is useful to reduce the head oscillations.

Overall, this study contributes to our understanding of how anguilliform fish obtain super swimming performance by adjusting the body stiffness and the muscle actuations, and it is also helpful to offer a general guidance to design the robotic fish with variable stiffness.

## Figures and Tables

**Figure 1 biomimetics-08-00263-f001:**
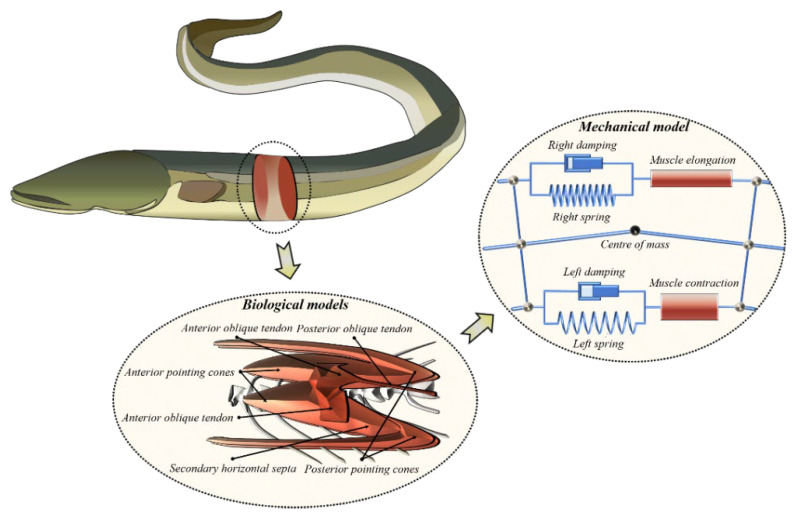
Dynamic model of anguilliform fish described by the redundantly actuated planar serial-parallel structures.

**Figure 2 biomimetics-08-00263-f002:**
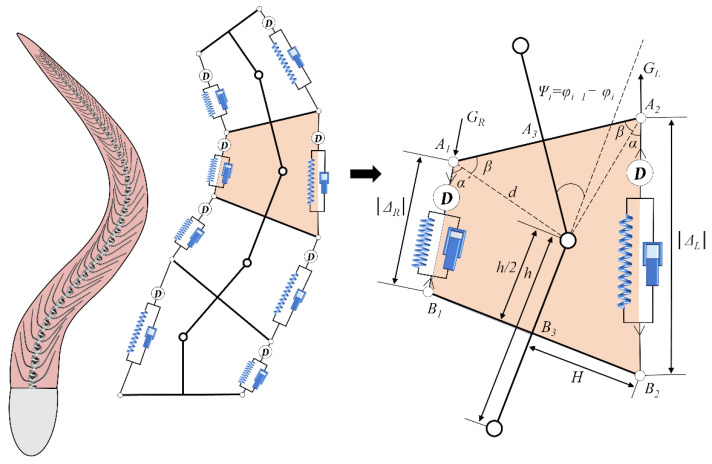
Schematic of the redundantly planar serial-parallel structures with muscle actuation.

**Figure 3 biomimetics-08-00263-f003:**
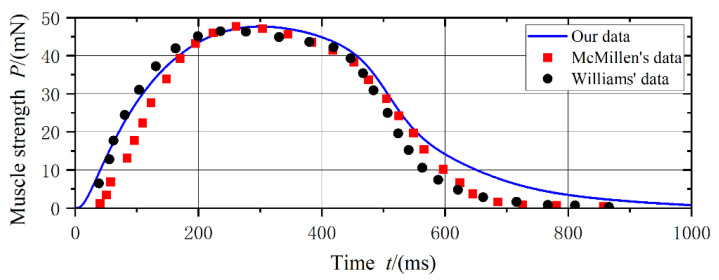
The intrinsic shape of actuation force generated by the muscle model of calcium ion.

**Figure 4 biomimetics-08-00263-f004:**
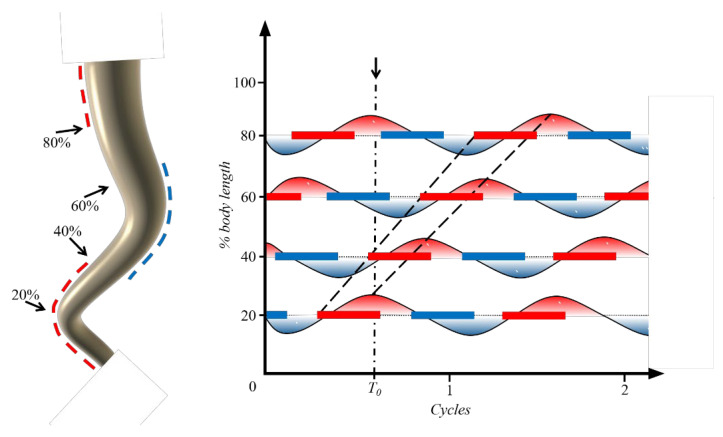
Sketch of the electromyograms (EMG) signal passage and the maximum convex (red lines) and concave (blue lines) curvature of bending body during swimming cycles.

**Figure 5 biomimetics-08-00263-f005:**
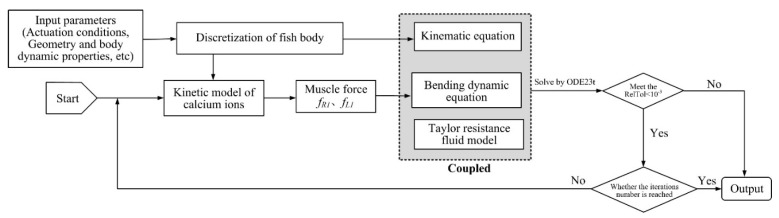
Solution flow chart to solve the dynamic model of anguilliform fish.

**Figure 6 biomimetics-08-00263-f006:**
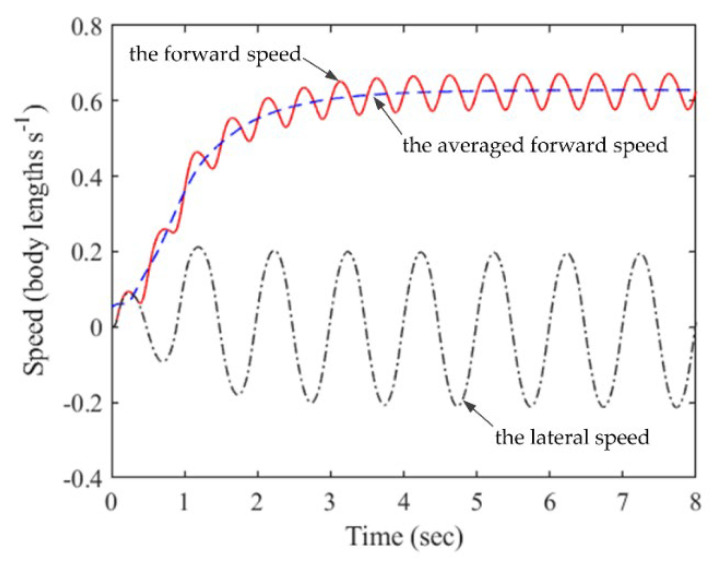
Time history of the lateral and forward velocities of anguilliform fish.

**Figure 7 biomimetics-08-00263-f007:**
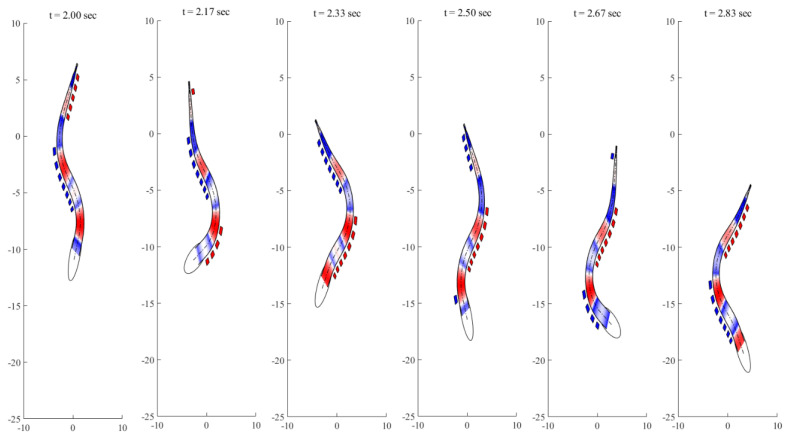
Undulatory motions of swimming fish during one tail-beat cycle (Red region: the active moment in fish body; Blue region: the positive moment in fish body; Red dashed lines on one side of body: the positive curvature region; Blue dashed lines near one side of body: negative curvature region).

**Figure 8 biomimetics-08-00263-f008:**
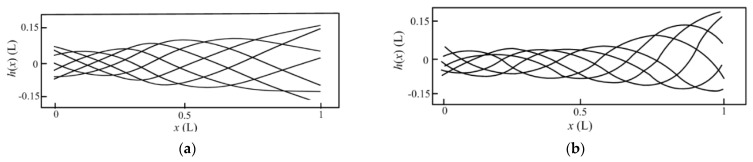
The undulatory motions of anguilliform fish in one tail-beat period. (**a**) Centerlines of the fish model. (**b**) Tracings from real lamprey fish (reproduced from [32]).

**Figure 9 biomimetics-08-00263-f009:**
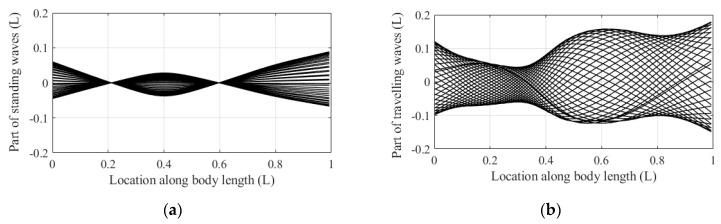
The decomposed parts of fish midline motions: (**a**) The part of pure standing wave; (**b**) The part of pure traveling wave.

**Figure 10 biomimetics-08-00263-f010:**
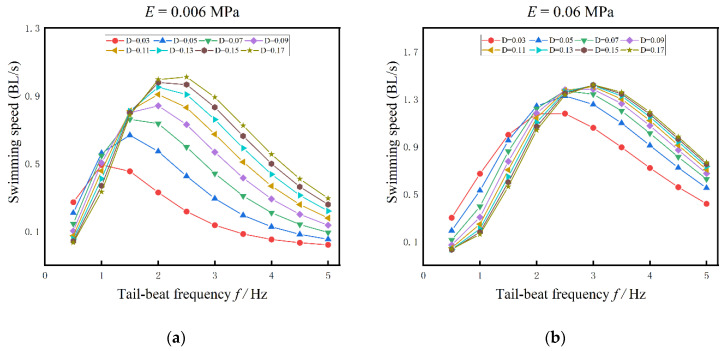
The influences of tail-beat frequency on swimming speed under different muscle actuation amplitudes. (**a**) E = 0.006 MPa (**b**) E = 0.06 MPa (**c**) E = 0.6 MPa (**d**) E = 1 MPa.

**Figure 11 biomimetics-08-00263-f011:**
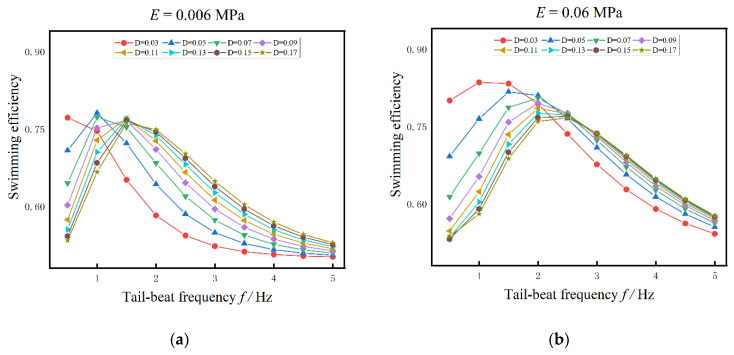
The influences of tail-beat frequency on swimming efficiency under different muscle actuation amplitudes. (**a**) E = 0.006 MPa (**b**) E = 0.06 MPa (**c**) E = 0.6 MPa (**d**) E = 1 MPa.

**Figure 12 biomimetics-08-00263-f012:**
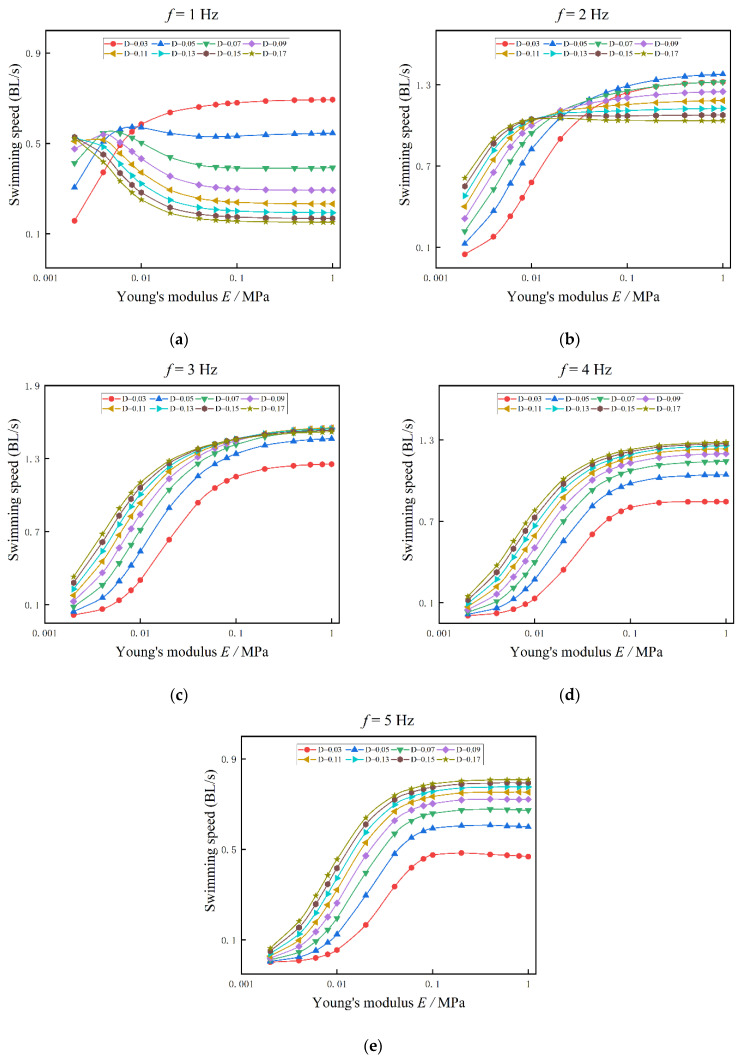
The influences of Young’s modulus on swimming speed under different muscle actuation amplitudes. (**a**) f = 1 Hz (**b**) f = 2 Hz (**c**) f = 3 Hz (**d**) f = 4 Hz (**e**) f = 5 Hz.

**Figure 13 biomimetics-08-00263-f013:**
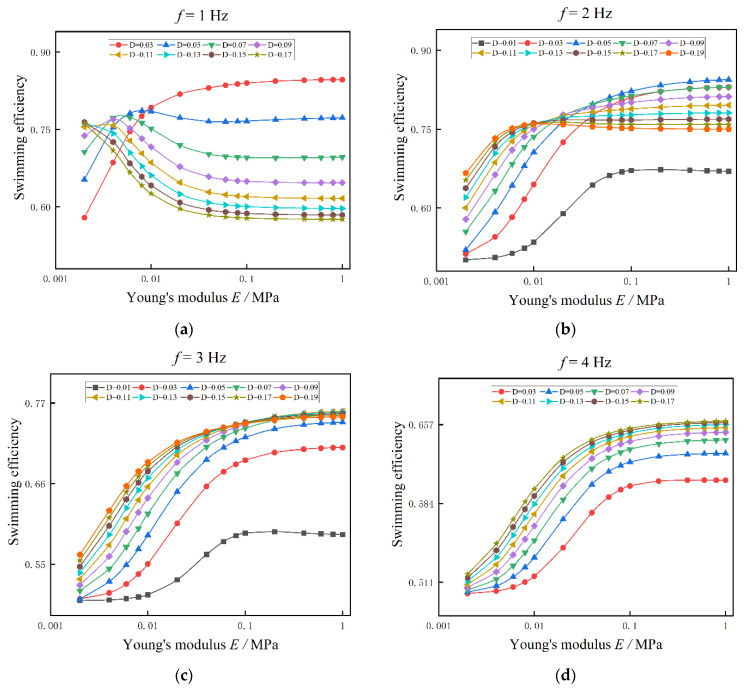
The influences of Young’s modulus on swimming efficiency under different muscle actuation amplitudes. (**a**) f = 1 Hz (**b**) f = 2 Hz (**c**) f = 3 Hz (**d**) f = 4 Hz (**e**) f = 5 Hz.

**Figure 14 biomimetics-08-00263-f014:**
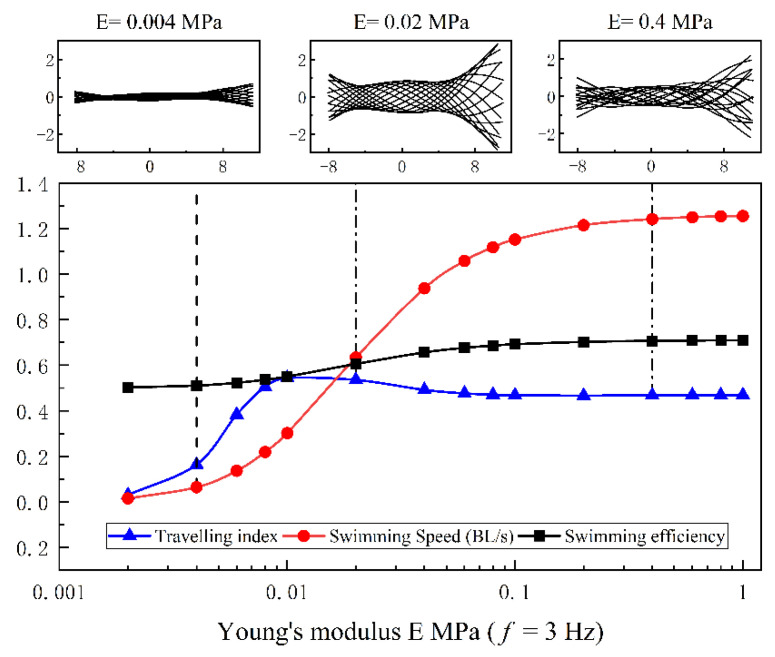
The traveling index of midline motions varied with Young’s modulus of fish body.

**Figure 15 biomimetics-08-00263-f015:**
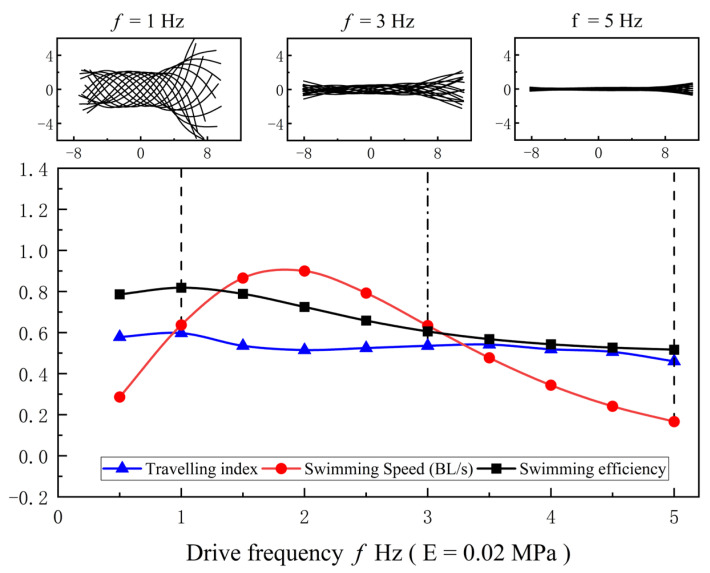
The traveling index of midline motions varied with the tail-beat frequency.

**Figure 16 biomimetics-08-00263-f016:**
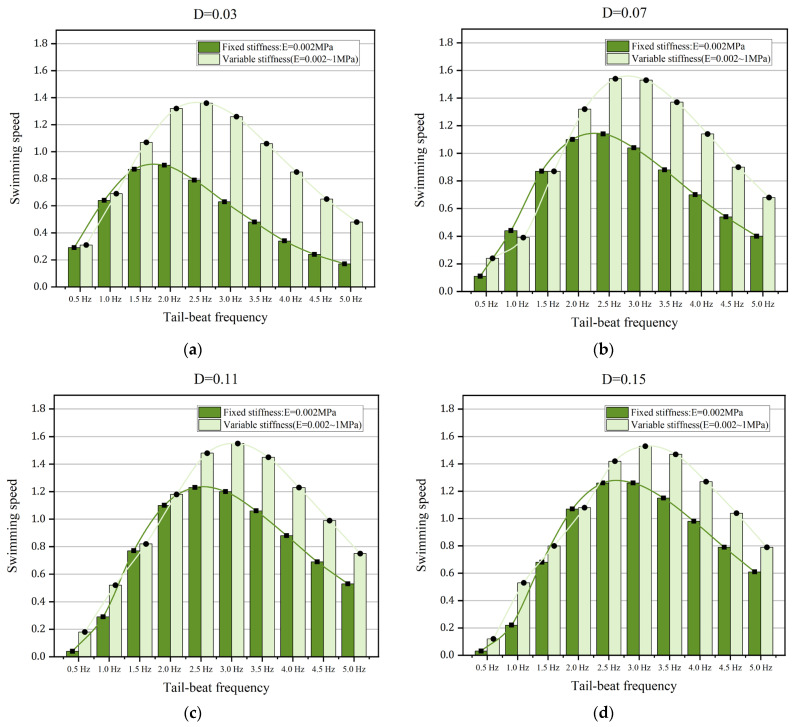
The comparisons of the swimming speed of anguilliform fish with fixed or variable body stiffness. (**a**) D = 0.03 (**b**) D = 0.07 (**c**) D = 0.11 (**d**) D = 0.15.

**Figure 17 biomimetics-08-00263-f017:**
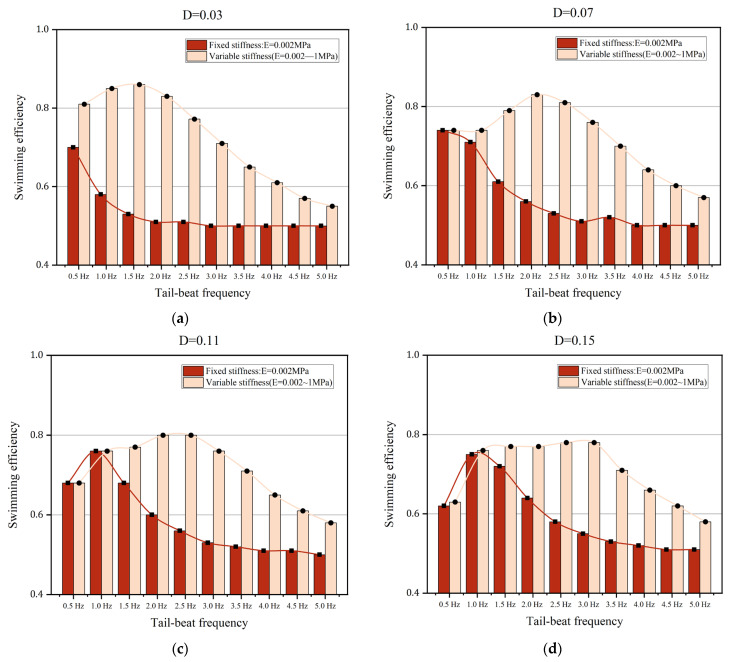
The comparisons of the swimming efficiency of anguilliform fish with fixed or variable body stiffness. (**a**) D = 0.03 (**b**) D = 0.07 (**c**) D = 0.11 (**d**) D = 0.15.

**Figure 18 biomimetics-08-00263-f018:**
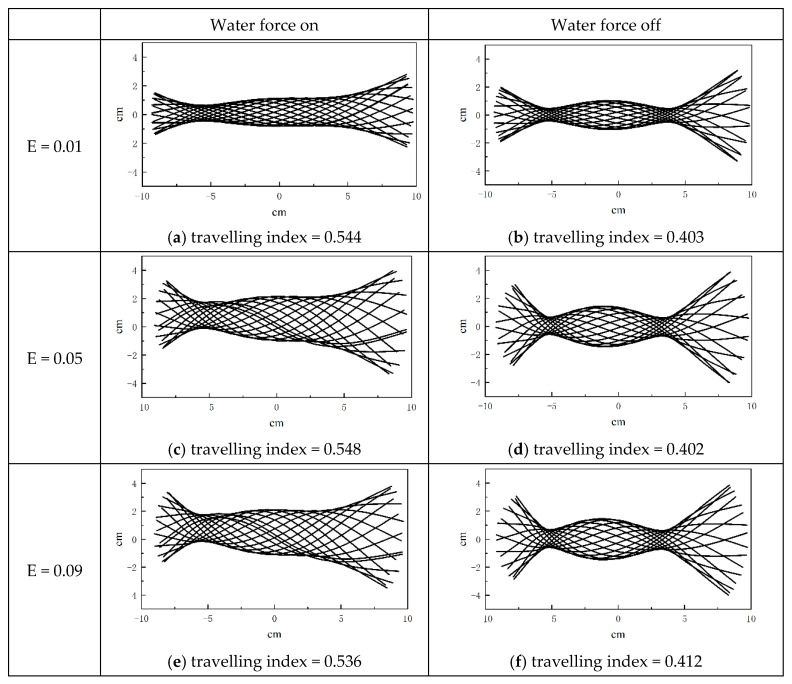
Successive snapshots of fish midline motions. The left and right panels show the motions with and without fluid forces.

**Table 1 biomimetics-08-00263-t001:** The variables of fish dynamic model.

Parameters	Symbols	Values
Length of fish body	L	20 cm
Body segment number	N	40
Water density	$\rho_{f}$	1.0 × 10^3^ kg/m^3^
Viscosity coefficients of water	$\mu_{f}$	1.0 mPa·s
Density of fish body	ρ	1.0 × 10^3^ kg/m^3^
Young’s modulus of fish body	E	0.1 MPa
Viscosity coefficient of fish body	μ	3 kPa·s
Actuation frequency	f	1 Hz

## Data Availability

The numerical and experimental data used to support the findings of this study are included within the article.

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
