# Peer review of "Computational Study of Stiffness-Tuning Strategies in Anguilliform Fish"

_biomimetics, 2023, doi:10.3390/biomimetics8020263_

Round 1

Reviewer 1 Report

This is a computational study on how fish adjust their body stiffness to improve swimming thrust and efficiency. A musculo-skeletal model of an anguilliform fish is developed, simulating muscle activities and generating force. The study investigates the relationships between forward speed, swimming efficiency, and body stiffness. Results indicate that swimming speed and efficiency increase with tail-beat frequency until reaching a maximum, and are also influenced by muscle actuation amplitude. Fish exhibit a tendency to vary body stiffness for better swimming performance, particularly at high tail-beat frequencies or small muscle actuation amplitudes. The study also analyzes fish motions and discusses the coordination between muscle actuation, body stiffness, and tail-beat frequency. The paper concludes with the intention to further explore this mechanism and its implications.

The presented work clearly articulates certain results and discussions, which are of interest to the research community. The paper exhibits a well-written structure, and the conclusions drawn are well-supported. The technical description of the results and discussions in the paper is comprehensive and easily understandable. Based on this, I  recommend the publication of the paper.

Minor comments:

1)Some abbreviations used in the manuscript, such as "COD" in the abstract and "ODE" in the main text, are not defined prior to their usage.

2)In section 4.1, titled "Fish body with fixed body stiffness," four different Young's modulus values (0.006 MPa, 0.06 MPa, 0.6 MPa, and 1.0 MPa) are selected. However, in section 4.2.2, titled "Influences on the midline motions," the range of Young's modulus for the fish body is stated to be from 0.002 MPa to 1 MPa, with three specific values (0.004 MPa, 0.02 MPa, and 0.4 MPa) chosen to study the effects of variable stiffness on midline motions. This discrepancy raises the question of why there is a lack of consistency in the selection of Young's modulus values.

Good enough for publication.

Author Response

Response to comments from Reviewer 1:

The presented work clearly articulates certain results and discussions, which are of interest to the research community. The paper exhibits a well-written structure, and the conclusions drawn are well-supported. The technical description of the results and discussions in the paper is comprehensive and easily understandable. Based on this, I recommend the publication of the paper.

[Reply] Thanks very much for the positive comments. We appreciate the time and effort that the editors and the reviewers have dedicated to providing the valuable feedbacks on our manuscript. Following the Reviewer’s suggestions, we have revised the manuscript thoroughly. Below are our point-by-point responses.

Minor comments:

(1) Some abbreviations used in the manuscript, such as "COD" in the abstract and "ODE" in the main text, are not defined prior to their usage.

[Reply] Thank you for pointing out this problem. In the revised manuscript, all the abbreviations are defined prior to the usage. Following the suggestion, we added the following clarifications: In the abstract (Page 1, Line 19-20): “Furthermore, the midline motions of anguilliform fish are analyzed by the complex orthogonal decomposition (COD) method, …” In the main text (Page 6, Line 233-234): “If the entire fish body is actuated, 6(N-1) first order ordinary differential equations (ODEs) describe the muscle forces in the N-link chain, and…”

(2) In section 4.1, titled "Fish body with fixed body stiffness," four different Young's modulus values (0.006 MPa, 0.06 MPa, 0.6 MPa, and 1.0 MPa) are selected. However, in section 4.2.2, titled "Influences on the midline motions," the range of Young's modulus for the fish body is stated to be from 0.002 MPa to 1 MPa, with three specific values (0.004 MPa, 0.02 MPa, and 0.4 MPa) chosen to study the effects of variable stiffness on midline motions. This discrepancy raises the question of why there is a lack of consistency in the selection of Young's modulus values.

[Reply] Thank you very much for your comments. In present study, the Young's modulus of fish body is varied from 0.002 MPa to 1 MPa, and all the cases with various Young's modulus, tail-beat frequency, muscle actuation coefficients have been simulated. Based on the results, we selected the appropriate cases to demonstrate the influences of stiffness-tuning strategies. In Section 4.2.2, the Young's modulus of fish body is ranged from 0.002 MPa to 1 MPa (as shown in Fig.14), and the changing pattern of swimming speed can be divided into three zones: low-speed zone, speed-increasing zone and high-speed zone. In each zone, one typical Young’s modulus, 0.004 MPa, 0.02 MPa and 0.4 MPa, are selected to study the influences of variable stiffness on the midline motions. If we choose the similar values in Section 4.1 (i.e., 0.006 MPa, 0.06 MPa and 0.6 MPa), the results of midline motions are similar when the Young’s modulus is 0.06 MPa and 0.6 MPa, respectively, due to their similar swimming performance in Fig.14. Following the suggestion, we added the following clarifications (Page 16, Lines 442-445): “The changing pattern of swimming speed can be divided into three zones: low-speed zone, speed-increasing zone and high-speed zone. In each zone, the typical Young’s modulus of fish body, 0.004 MPa, 0.02 MPa and 0.4 MPa, are selected to demonstrate the influences of variable stiffness on the midline motions.”

Reviewer 2 Report

The work "Computational Study of Stiffness-tuning Strategies in Anguilliform Fish" by Cui and Zhang presents a novel simulation model that analyze fish's swimming speed and efficiency with respect to body stiffness, tail-beating frequencies and muscle actuations. This work presents innovative insights that future robotic swimmers can adapt to improve working efficiency. The results are clearly demonstrated with sufficient data to back up. I suggest for publication after a few minor edits.

1. In the section of the "Introduction", the author did not address the meaning of the work. Why we are trying to study the body-stiffness tuning strategy for fish to swim faster? Any potential benefits if we know the reasons behind these strategies? Please address

2. "Line 71-74" Please explain the novel strategies the authors used to build the model. What is the difference between the introduced model compared to previous results

3. "Line 80-81“. Please add "at the moment" as we could potentially "how fish use active muscle tensioning to tune their body stiffness to improve thrust and/or efficiency in real-time" in future.

Author Response

Response to comments from Reviewer 2:

The work "Computational Study of Stiffness-tuning Strategies in Anguilliform Fish" by Cui and Zhang presents a novel simulation model that analyze fish's swimming speed and efficiency with respect to body stiffness, tail-beating frequencies and muscle actuations. This work presents innovative insights that future robotic swimmers can adapt to improve working efficiency. The results are clearly demonstrated with sufficient data to back up. I suggest for publication after a few minor edits.

[Reply] We thank Reviewer 2 for the helpful comments and suggestions, and appreciate the time and effort that the editors and the reviewers have dedicated to providing the valuable feedbacks on our manuscript. Following the Reviewer’s suggestions, we have revised the manuscript thoroughly. Below are our point-by-point responses.

Minor comments:

(1) In the section of the "Introduction", the author did not address the meaning of the work. Why we are trying to study the body-stiffness tuning strategy for fish to swim faster? Any potential benefits if we know the reasons behind these strategies? Please address.

[Reply] Revised as suggested. Some clarifications are added in the revised manuscripts in Page 2, Lines 86-93: “The stiffness-tuning strategies in anguilliform fish can be implemented in next generation fish-like robots or underwater vehicles, with an immediate improvement in forward speed and propulsive efficiency. The stiffness-tuning mechanisms and automatic stiffness control based on various swimming conditions are also promising applications. Moreover, while the stiffness-tuning strategies are poised to improve fish-like robots, they would offer insights into new studies of live fish locomotion, with the requirements of new challenging experiments, such as invasive surgery with implanting transducers and electrodes into fish body, vivo stiffness measurements of various fish species, etc.”

(2) "Line 71-74" Please explain the novel strategies the authors used to build the model. What is the difference between the introduced model compared to previous results.

[Reply] Thank you for your valuable comment. The differences between our fish model and previous model (proposed by McMillen et al.) include: 1) the redundantly actuated planar series-parallel structure is proposed to demonstrate the biological structure of fish body, in which the stiffness can be adjusted as needed. In our previous studies, we also focus on the stiffness-tuning mechanisms, such as the design of active and passive stiffness in the planar series-parallel structure, see our previous publications [1-2]. 2) The influences of various body stiffness on swimming performance is explored in present study, which is not studied in the works of McMillen et al. Following the suggestions, some clarifications are added in the revision (Page 2, Lines 72-75). “Different from the previous model, the redundantly actuated planar series-parallel structure is adopted to demonstrate the biological structure of fish body, in which the stiffness can be adjusted as needed.” [1] Li K, Jiang H, Cui Z. Design for solving negative stiffness problem of redundant planar rotational parallel mechanisms [J]. International Journal of Robotics and Automation, 2019, 34(1): 78-83. [2] Chen B, Cui Z, Jiang H. Producing negative active stiffness in redundantly actuated planar rotational parallel mechanisms [J]. Mechanism and Machine Theory, 2018, 128: 336-348.

(3) "Line 80-81". Please add "at the moment" as we could potentially "how fish use active muscle tensioning to tune their body stiffness to improve thrust and/or efficiency in real-time" in future.

[Reply] Revised as suggested. We updated the manuscript in Page 2, Lines 82-85: “Although how fish use active muscle tensioning to tune their body stiffness to improve thrust and/or efficiency in real-time cannot be fully explained at the moment, we try to throw light on this problem in present study.”
